# Molecular and Cellular Involvement in CIPN

**DOI:** 10.3390/biomedicines12040751

**Published:** 2024-03-28

**Authors:** Housem Kacem, Annamaria Cimini, Michele d’Angelo, Vanessa Castelli

**Affiliations:** Department of Life, Health and Environmental Sciences, University of L’Aquila, 67100 L’Aquila, Italy; houseneddine.kacembenhajmbarek@graduate.univaq.it (H.K.); annamaria.cimini@univaq.it (A.C.); vanessa.castelli@univaq.it (V.C.)

**Keywords:** neuropathy, chemotherapy, mechanisms, neuroinflammation, channels

## Abstract

Many anti-cancer drugs, such as taxanes, platinum compounds, vinca alkaloids, and proteasome inhibitors, can cause chemotherapy-induced peripheral neuropathy (CIPN). CIPN is a frequent and harmful side effect that affects the sensory, motor, and autonomic nerves, leading to pain, numbness, tingling, weakness, and reduced quality of life. The causes of CIPN are not fully known, but they involve direct nerve damage, oxidative stress, inflammation, DNA damage, microtubule dysfunction, and altered ion channel activity. CIPN is also affected by genetic, epigenetic, and environmental factors that modulate the risk and intensity of nerve damage. Currently, there are no effective treatments or prevention methods for CIPN, and symptom management is mostly symptomatic and palliative. Therefore, there is a high demand for better understanding of the cellular and molecular mechanisms involved in CIPN, as well as the development of new biomarkers and therapeutic targets. This review gives an overview of the current knowledge and challenges in the field of CIPN, focusing on the biological and molecular mechanisms underlying this disorder.

## 1. Introduction

Chemotherapy-induced peripheral neuropathy (CIPN) frequently emerges as an adverse outcome linked with various cancer therapies, such as taxanes, platinum compounds, vinca alkaloids, and proteasome inhibitors [1,2]. It manifests through sensations of paresthesia, dysesthesia, and frequent discomfort, predominantly observed in the upper and lower limbs, like hands and feet [3,4,5]. Patients frequently describe sensory experiences such as loss of sensation and pins and needles, succeeded by feelings like scorching, shooting, pulsing, and itching. Each of these symptoms is linked to a reduced capacity to discern tactile and pinprick sensations, alongside compromised sensorimotor functions [1]. The severity of these sensory alterations might require a dosage reduction in the therapy provided to the patient. About 40% of individuals receiving cisplatin or paclitaxel therapy will encounter CIPN, whereas an estimated 80% of those undergoing oxaliplatin treatment report heightened sensitivity to cold [6]. The initiation of CIPN typically correlates with the dosage, particularly the amount of medication given and the frequency of administrations, as well as diverse risk factors like diabetes or smoking. The latest estimate of cancer prevalence stands at 18.1 million new cases annually [7]. With the availability of more efficacious targeted cancer therapies, cancer survival is increasing in wealthy nations, as demonstrated by the 27% decline in the total cancer mortality rate in the United States from 1991 to 2016 [8,9,10]. Nonetheless, CIPN poses a frequent and formidable challenge arising from various commonly utilized antineoplastic agents [11], potentially resulting in extended infusion durations, dosage adjustments, or premature discontinuation of chemotherapy, all of which could detrimentally affect both treatment effectiveness and patient survival [12]. A review of randomized controlled trials and cohort studies revealed that roughly half of patients develop CIPN throughout their treatment regimen [13]. There is no universally accepted method for evaluating CIPN, as studies use different clinical tools with diverse primary outcome measures [14]. Certainly, the definition and detection of subclinical nerve damage and motor involvement remain inadequate using standardized clinical tools [15].

Precise comparisons regarding the prevalence, incidence, prevention, and treatment of CIPN pose challenges. Moreover, significant differences exist between patient-reported and clinician-reported neurotoxicity. For instance, in the ICON7 trial, clinicians noted CIPN in 28% of patients, whereas 67% of patients reported experiencing ‘quite a bit’ or ‘very much’ tingling or numbness. Notably, there was low concordance between patients and clinicians (κ = 0.236, 95% confidence interval, 0.177–0.296, *p* < 0.001) [16].

Chemotherapeutic drugs induce neurotoxic effects via diverse pathways, leading primarily to a symmetric sensory or sensorimotor neuropathy with a length-dependent pattern, alongside autonomic dysfunction [17,18]. Distinct neuropathic syndromes due to the administration of chemotherapeutic agents can be identified, each characterized by unique clinical features and progression over time [19,20]. CIPN may emerge or continue to deteriorate several months following cessation of treatment, a phenomenon referred to as ’coasting’. Approximately 68% of patients encounter CIPN post-chemotherapy, with its persistence observed in roughly one-third of patients beyond the 6-month mark [12].

As more efficacious chemotherapeutic regimens are available, cancer cure rates and long-term cancer survival will continue to increase, along with CIPN [21]. Hence, it is crucial to devise efficient approaches for promptly identifying, preventing, and managing CIPN more effectively. In the context of CIPN, non-pharmacological therapeutic strategies play a key role in alleviating symptoms and enhancing patient well-being. These approaches may encompass pain management techniques, physiotherapy, occupational interventions, as well as psychotherapeutic interventions aimed at mitigating the emotional burden associated with neuropathy [22]. Targeted therapies focusing on molecular targets emerge as an intriguing alternative for CIPN treatment. With specific attention directed toward the molecular mechanisms underlying neurotoxicity stemming from chemotherapy, these therapeutic modalities can serve a preventative or mitigative role against neuronal damage and, consequently, improve the functionality of compromised peripheral nerves [23]. Monoclonal antibodies (mAbs) emerge as an effective therapeutic strategy in managing CIPN. Through targeted binding to specific molecules involved in neurotoxic processes, monoclonal antibodies can provide protection to peripheral nerves from chemotherapy-induced injuries, thereby mitigating neuropathic symptoms and enhancing the quality of life for patients with oncological conditions [24].

This review aims to explore the complex molecular and cellular mechanisms behind the development of CIPN. Through this analysis, we strive to provide a critical and in-depth understanding of the intricate molecular and cellular interactions that lead to the manifestation of CIPN. The primary objective is to identify and comprehend the biochemical processes and cellular alterations due to exposure to chemotherapy treatments that cause CIPN. This will help deepen the understanding of the molecular mechanisms underlying CIPN, seeking to pinpoint potential signaling pathways, cellular modifications, and interactions that may be involved in the onset of this complication. This knowledge could ultimately guide the development of more targeted therapeutic approaches and effective preventive strategies to enhance the quality of life for patients undergoing chemotherapy treatments.

## 2. Underlying Mechanisms of CIPN

The underlying mechanisms of CIPN are not fully known, but they involve direct nerve damage, oxidative stress, inflammation, DNA damage, and microtubule dysfunction. Altered ion channel transmission, along with alterations in intracellular signaling and structures, have been also implicated in CIPN [25]. Current studies suggest that there are also genetic predispositions that contribute to the development of this condition. In the next paragraphs, we will see the different mechanisms in detail.

### 2.1. Neurophysiological Mechanisms of CIPN

Voltage-gated ion channels are membrane proteins that regulate the flow of ions across the cell membrane in response to changes in the membrane potential. They play important roles in various physiological processes, such as neuronal excitability, synaptic transmission, muscle contraction, and hormone secretion [26]. One of the mechanisms of CIPN is the alteration of voltage-gated ion channel function and expression in peripheral sensory neurons, especially the small-diameter nociceptive neurons that transmit pain signals. Different types of voltage-gated ion channels, such as sodium, potassium, calcium, and chloride channels, may be affected by different chemotherapy drugs, depending on their chemical structure, mode of action, dose, duration, and individual susceptibility [27]. Chemotherapeutics are able to increase the expression and activity of voltage-gated sodium channels (VGSCs), especially Nav1.7 and Nav1.8, which mediate the initiation, amplification, and conduction of action potentials in nociceptive neurons. This leads to hyperexcitability and spontaneous firing of these neurons, resulting in increased pain sensitivity and neuropathic pain [27]. It has been demonstrated that dexpramipexole, a specific inhibitor of NaV1.8, induced substantial analgesic effects in a mouse model of oxaliplatin-induced peripheral neuropathy [28]. Additionally, other NaV1.8 blockers have exhibited efficacy in treating neuropathic pain, with phase II trials currently in progress.

A simulation analysis suggested that reductions in the activity of voltage-gated K+ channels (K_V_) induced by oxaliplatin and enhancements in sodium channel activity can account for the documented nociceptor hyperexcitability [29,30]. Chemotherapeutics act mainly on the expression and activity of Kv7 and Kv11, which mediate the repolarization and stabilization of the membrane potential in nociceptive neurons, leading to depolarization and increased excitability of these neurons, resulting in increased pain sensitivity and neuropathic pain [31].

Voltage-gated calcium channels also have a crucial role in neuronal excitability and CIPN development and maintenance [32]. For example, paclitaxel can increase the expression and activity of low-voltage-activated calcium channels (T-type; Ca v 3.2) in dorsal root ganglion neurons, which are the primary sensory neurons of the peripheral nervous system. This can lead to increased neuronal excitability, spontaneous firing, and ectopic activity, resulting in abnormal pain sensation and transmission [33,34].

Therefore, voltage-gated ion channels are potential targets for the prevention and treatment of CIPN. It is notable that blockers of voltage-gated sodium channels, such as the antiepileptic medication carbamazepine, have shown certain degrees of effectiveness in managing neuropathy in individuals, although not all clinical investigations have corroborated the efficacy of this strategy [35,36]. Other pharmacological agents that modulate voltage-gated ion channels have been tested in preclinical and clinical studies for CIPN, such as lidocaine, carbamazepine, lamotrigine, retigabine, gabapentin, pregabalin, and lacosamide [37,38,39]. However, the results have been inconclusive, and none of these drugs have been approved by the FDA for CIPN. More research is needed to identify the optimal dose, timing, and combination of these drugs, as well as to discover new and more selective modulators of voltage-gated ion channels for CIPN.

### 2.2. Modifications in Transient Receptor Potential Channels in CIPN

Transient Receptor Potential (TRP) channels are a family of ion channels that are expressed in sensory neurons and can be activated by various stimuli, such as temperature, chemicals, and mechanical forces. Some TRP channels, such as TRPV1, TRPV4 (vanilloid), TRPA1 (ankyrin), and TRPM8 (melastatin), expressed on the plasma membrane of primary sensory neurons, have been implicated in the development of CIPN [40].

Indeed, they may contribute to CIPN by mediating the neurotoxic effects of chemotherapy drugs on sensory neurons, or by modulating the sensitivity and excitability of nociceptors in response to chemotherapy-induced damage by different mechanism. For instance, TRPV1, which is activated by heat and capsaicin, has been shown to be involved in paclitaxel-, cisplatin- and oxaliplatin-induced neuropathy, possibly mediating axonal degeneration by increasing calcium influx and oxidative stress in sensory neurons exposed to these agents [41,42,43,44].

Regarding TRPV4, which is usually activated by heat and mechanical stimuli, it has been implicated in taxane-induced neuropathy [43]; indeed, TRPV4 antagonists or knock-out mice attenuate pain and nerve injury.

On the other hand, TRPA1 is activated by cold and reactive electrophiles. Studies demonstrated its involvement in CIPN, and specifically platinum derivative-induced cold [40] allodynia and mechanical hypersensitivity [40], possibly due to increased oxidative stress [45]. Finally, TRPM8, activated by cold and menthol, has been associated with CIPN. The role of this channel in CIPN is not fully understood, but some studies have suggested that it may have both protective and detrimental effects in cancer progression and CIPN control. On the one hand, TRPM8 activation may help in preventing or reducing CIPN by enhancing the survival and regeneration of sensory neurons and by modulating pain perception and inflammation. On the other hand, TRPM8 activation may also worsen CIPN by increasing the sensitivity and excitability of sensory neurons and by promoting the growth and invasion of cancer cells (extensively reviewed in [46]). Therefore, more research is needed to clarify its mechanisms and therapeutic implications.

Overall, TRP channels represent potential targets for the prevention and treatment of CIPN, as they may offer a way to modulate the sensory input from damaged nerves and reduce the chronic pain associated with CIPN. However, more research is needed to elucidate the exact role of each TRP channel in different models of CIPN and to evaluate the safety and efficacy of TRP channel modulators in clinical trials.

### 2.3. Neuroinflammation in CIPN

Recently, the scientific community has been focusing on the probable role of neuroinflammation in the pathophysiology of CIPN and has highlighted the connection between these two mechanisms. Indeed, chemotherapeutic treatments cause a deregulation of the molecular mechanisms of immune cells, thus leading to the development of neuroinflammation [47]. The precise immunological pathways targeted by chemotherapy remain elusive, yielding conflicting outcomes. Numerous investigations showcase the immunosuppressive nature of chemotherapy owing to its cytotoxic effects [48]. Conversely, several pieces of evidence indicate that certain chemotherapeutic agents elicit immune system stimulation. For example, some results obtained from animal models show the infiltration of leukocytes in the nervous system. The same data demonstrate how peripheral glia can be more or less activated depending on the type of chemotherapy used, the dose, and treatment times [49]. In a 2017 study, male C57BL/6J mice were treated with paclitaxel and oxaliplatin for one week, and a significant increase in circulating populations of CD4+ and CD8+ T cells in the periphery was reported, suggesting a more aggressive immune response [50]. At DRG level, the treatment with paclitaxel caused an upregulation of the activating transcription factor 3 (ATF-3), a typical marker of nerve damage [50,51].

ATF-3 is a protein involved in gene expression regulation. This protein is known to be involved in various cellular processes, including stress response and inflammation. Its expression can be triggered by a variety of stimuli, including oxidative stress, inflammation, and DNA damage. In the context of CIPN, ATF-3 expression in Schwann cells has been linked to the peripheral nerve response to chemotherapy-induced damage. Recent research indicates that chemotherapy treatments induce an increase in ATF-3 levels and alterations at the mitochondrial and endoplasmic reticulum levels in satellite glial cells of DRG [48].

Upon platinum derivatives, increased expression and secretion of pro-inflammatory cytokines and chemokines, such as TNF-α, IL-1β, IL-6, CCL2, and IL-1β, were observed [47]. After peripheral nerve damage, microglial cells release TNF-α and IL-1β, stimulating astrocytes. Once activated, these release C-C Motif Chemokine Ligand 2 (CCL2), which appears to be involved in the pathophysiology of neuropathic pain [48]. IL-1β activates nociceptive neurons through binding to the IL-1 receptor, activating the mitogen-activated protein kinase (MAPK) pathway, thereby increasing neuronal sensitivity. Furthermore, IL-1β and TNF-α regulate the phosphorylation of N-Methyl-D-aspartic acid (NMDA) receptor subunits, NR2B and NR1, leading to an increase in NMDA-induced currents, suggesting enhanced glutamatergic synaptic transmission. Confirming the key role of TNF-α and IL-1β in the modulation of CIPN, recent studies carried out on patients have demonstrated the presence of elevated levels of these cytokines in spinal astrocytes [52].

The use of vincristine also appears to impact the release of various cytokines, including the aforementioned TNF-α, IL-6, and IL-1β, which contribute to inducing a state of neuroinflammation. Recent studies have further highlighted that treatment with this chemotherapeutic agent results in a decrease in interleukin 10, which possesses anti-inflammatory properties [53]. Another pathway that appears to play a key role in CIPN is NF-κB and its binding to its receptor CXCR2 [54]. Some studies conducted on animals treated with vincristine confirmed the presence of elevated levels of CXCL1 induced by NF-κB. The crucial role of this factor was confirmed by the increase in TNF-α and the decrease in IL-10 levels [55]. A recent study conducted both in vitro and in vivo elucidated the mechanisms by which paclitaxel engages and activates the receptor 1 of complement component 5a (C5aR1), thereby initiating the NF-κB/P38 pathway. Remarkably, the inhibition of this receptor was shown to effectively mitigate the onset of neuroinflammation and ameliorate the CIPN condition, restoring the aberrant electrical activity of the DRG [41].

In addition to the cytokines described, chemokines also appear to play a crucial role in activating the inflammatory condition involved in CIPN. Activated glial cells play a key role in maintaining chronic pain through mechanisms involving synaptic remodeling and hyperexcitability of pain pathways. Microglia, in response to tissue damage or nerve injury, assume distinct phenotypes (M1 and M2), influencing inflammation and the immune response [56]. Additionally, astrocytes closely communicate with microglia and neurons, releasing mediators such as CCL2, CXCL1, and CXCL13 that enhance chronic pain by promoting synaptic transmission [57]. Another relevant aspect concerns the role of chemokines in the activation and infiltration of immune cells, such as macrophages and glial cells, in CIPN. It has been observed that chemotherapy increases the expression of chemokines, such as CCL2 and CX3CL1, in sensory neurons, influencing the inflammatory response [48]. In particular, the CCL2/CCR2 signaling pathway appears to play a key role in recruiting and activating immune cells, contributing to hypersensitivity and pain [58].

Another chemokine involved in CIPN is IL-8, in particular in paclitaxel- and oxaliplatin-induced neuropathy, as demonstrated by recent in vivo and in vitro studies. In this study, to confirm the crucial role of this cytokine, selective inhibitors of IL-8 receptors in combination with the antineoplastic were used and significantly attenuated CIPN condition [59,60].

In summary, neuroinflammation plays a significant role in the pathophysiology of CIPN, interacting intricately with immune responses. Chemotherapy alters the molecular mechanisms of immune cells, promoting neuroinflammation. However, the specific immunological pathways targeted by chemotherapy remain partly unclear, with conflicting results. Understanding the communication between glial and neuronal cells is essential to grasp the pathogenesis of CIPN and to identify potential therapeutic interventions. Chemokines, in particular, represent promising targets for treating neuropathic pain associated with CIPN. Further studies to elucidate the mechanisms underlying neuroinflammation and to identify new therapies are necessary to significantly improve the management of CIPN and the quality of life of affected patients.

### 2.4. Oxidative Stress and Mitochondrial Dysfunction in CIPN

It has been reported that chemotherapeutic agents led to mitochondrial free radical production and increased oxidative stress. Consequently, sensory neurons and peripheral nerves are compromised due to mitochondrial injury, demyelination, microtubular damage, mitophagy and apoptosis [61]. In addition, taxanes, vinca alkaloids, and platinum compounds cause axonal mitotoxicity, affecting nerve function [62,63].

In CIPN, mitochondrial dysfunction disrupts calcium signaling pathways within neurons, depending on the chemotherapeutic agent used. For instance, paclitaxel causes the opening of the mitochondrial permeability transition pore in axons, resulting in mitochondrial membrane potential loss, increased ROS, reduced ATP levels, calcium release, and mitochondrial swelling. Platinum compounds form adducts with mitochondrial DNA, inhibiting replication, disrupting transcription, and causing morphological abnormalities within mitochondria in DRG neurons, leading to gradual altered metabolism and energy failure [64,65]. The proteasome inhibitor bortezomib modifies mitochondrial respiratory chain and mitochondrial calcium homeostasis, inducing mitochondrial dysfunction and oxidative stress [66]. Altered expression of genes controlling mitochondrial functionality and those involved in AMPK-dependent signaling (responsible for cellular ATP supply) is observed in patients with bortezomib-induced peripheral neuropathy [67].

Additionally, paclitaxel can elevate mitochondrial ROS production, heightening TRPA1 channel sensitivity to amplify thermal sensitivity in rodents. As previously noted, ROS discharged from mitochondria can subsequently trigger apoptotic and proinflammatory pathways, exacerbating chronic CIPN [68]. Further evidence supporting ROS involvement in CIPN is the discovery that the administration of ROS scavengers mitigates paclitaxel-induced mechanical hyperalgesia [69,70]. Translational studies in rodents have suggested peroxynitrite, a potent oxidant and nitrating agent generated by ROS, as a potential therapeutic target to alleviate CIPN [71].

Moreover, chemotherapeutics can interfere with the antioxidant defense of sensory neurons, resulting in increased ROS production, damage of various cellular components, lipid peroxidation, mitochondrial membrane depolarization, and reduced ATP synthesis [72]. These changes can impair the neuronal function and survival, leading to the typical symptoms of CIPN. For instance, bortezomib and cisplatin have been shown to induce damage to organelles like lysosomes and endoplasmic reticulum within neurons and Schwann cells [73]. It has been proposed that numerous conventional chemotherapeutic agents act on microtubules within cancer cells to hinder cellular division. There is speculation that the attachment of these medications to neuronal microtubules might play a role in CIPN [74]. Impairment of axonal transport in peripheral nerves has been observed after treatment with vincristine and paclitaxel. Interestingly, neither oxaliplatin nor bortezomib directly binds to microtubules to induce cancer cell death, yet both still induce significant neuropathy in clinical populations [75]. Moreover, both bortezomib and oxaliplatin disrupt axonal transport similarly to antimicrotubule agents [76,77]. Hence, the inhibition of axonal transport by these latter two agents may occur through a mechanism distinct from microtubule interaction, suggesting that this alternate pathway might also play a role in axonal transport suppression by other chemotherapeutic drugs.

One of the main mechanisms underlying the development of CIPN is the damage to microtubules caused by antineoplastic agents [78]. Microtubules are essential structural elements within cells and integral components of the cytoskeleton, an intricate network of protein filaments that provide shape, support, and intracellular vesicle and organelle movement to cells and their internal components [79]. Antineoplastic drugs negatively impact the stability and proper functioning of microtubules, leading to alterations in their structure and compromising normal cellular activities [80]. This damage to microtubules in peripheral nerves can cause neuronal dysfunction and give rise to the typical symptoms of CIPN. Antineoplastic agents, including paclitaxel, taxanes, and vinca derivatives, constitute a class of essential drugs for cancer treatment, primarily operating through interference with microtubules. For example, paclitaxel is known for its ability to stabilize microtubules by blocking tubulin depolymerization, thereby halting cell division [81]. Oxaliplatin, classified as a platinum alkylating agent, exhibits a more intricate action within cells. Essentially, it forms crosslink adducts with DNA, a reaction that induces significant cellular damage. This interaction not only results in direct DNA damage but also activates crucial intracellular signaling pathways, leading to microtubule destabilization and subsequent cell cycle arrest [82].

Bortezomib is the first proteasome inhibitor approved by the FDA for clinical applications whose mechanism of action is the inhibition of the proteasome-ubiquitination system [83].

Malacrida et al. demonstrated for the first time the direct interaction of this drug with microtubules, suggesting that this may be the cause of its neurotoxic effect [66].

This drug prevents the degradation of key regulatory proteins within cells, thereby influencing microtubule stability regulation and cell proliferation [66]. Finally, vinca derivatives, including vincristine and vinblastine, exert their action by inhibiting microtubule polymerization. This mechanism interferes with the formation of the mitotic spindle, which in turn compromises proper chromosome segregation during cell division [84].

Vinca derivatives induce direct harm to peripheral neurons by disrupting the formation and stability of microtubules. Consequently, this disrupts intracellular transport, triggers inflammasome activation, and may lead to neuronal apoptosis, thereby exacerbating neuropathic symptoms [85].

Understanding the molecular and cellular mechanisms through which antineoplastic agents cause damage to microtubules and contribute to the onset of CIPN is crucial in the development of preventive and therapeutic strategies aimed at reducing or preventing this serious complication of antineoplastic therapy.

Therefore, oxidative stress and dysfunction in mitochondrial and other organelles are important factors in the pathogenesis of CIPN, and they could be potential targets for prevention and treatment strategies. Some of the proposed interventions include antioxidants, mitochondrial modulators, and anti-inflammatory agents [86,87,88]. However, more studies are necessary to establish the efficacy and safety of these approaches in a clinical setting.

### 2.5. Degeneration of Intraepidermal Nerve Fibers in CIPN

The combined impact of the described mechanisms appears to result in a reduction of intraepidermal nerve fibers (IENFs) and Meissner’s corpuscles (MC) in regions of the skin corresponding to the most pronounced symptoms of CIPN [89]. Meissner’s corpuscles, also known as tactile corpuscles, are nerve structures located in the papillary dermis, especially in touch-sensitive skin regions such as fingers, palms, and the soles of the feet [90]. Concentrated in areas highly sensitive to tactile stimuli, these corpuscles play a crucial role in tactile perception, being sensitive to changes in pressure and vibrations, contributing to tactile sensitivity and the discrimination of skin textures. Their presence is essential for accurate tactile perception, playing a significant role in the sensory response of the peripheral nervous system [91]. The observed loss of Meissner’s corpuscles may be associated with the reduced tactile perception in patients with CIPN [92]. Repetitive studies have underscored the correlation between the decline in intraepidermal nerve fibers (IENFs) and the severity of symptoms experienced by patients afflicted with chemotherapy-induced peripheral neuropathy (CIPN) caused by paclitaxel, oxaliplatin, and bortezomib [93,94]. Though the precise mechanisms driving this reduction in IENFs remain elusive, comparable observations are documented in other conditions associated with painful neuropathy, including HIV infection and diabetes [95,96]. It is important to highlight that safeguarding against the diminishment of IENFs via minocycline, a derivative of tetracycline known to alleviate neuroinflammation, confers shielding against neuropathy stemming from oxaliplatin and paclitaxel in animal models of CIPN [93,94]. Likewise, strategies targeting the suppression of CCL2, a pivotal chemokine orchestrating inflammatory reactions within the dorsal root ganglia following chemotherapy interventions, have effectively halted the behavioral manifestations of CIPN and prevented reduction in the density of distal intraepidermal nerve fibers (IENFs) in rats [97].

### 2.6. Genetic Evidence of CIPN

Recently, emerging evidence linked to higher susceptibility to chemotherapy-induced adverse effects suggested that genetics may influence the risk of developing CIPN, thereby facilitating tailored treatment personalization [98]. Several research endeavors, each dedicated to particular cancer types and associated treatments, have delved into the correlation between single-nucleotide polymorphisms (SNPs) and the predisposition to CIPN. A succinct examination of genes and SNPs suggested as plausible predictive markers for CIPN has been recently released [99]. Unfortunately, the results of these studies have not consistently demonstrated uniform reproducibility. As an example, numerous investigations have examined the GSTP1 gene. Prior studies indicated that variations in this gene were associated with survival rates in cancer patients, rendering it a promising contender for pharmacogenomic investigation [100]. Though some studies have found a connection between GSTP1 polymorphisms and CIPN, an equal number of studies have failed to observe this association, even when considering variables such as ethnicity, cancer subtype, and primary treatment [101,102]. However, this avenue of inquiry necessitates additional exploration, as many studies have prioritized genes associated with cancer rather than genes specifically implicated in neuropathy. Several investigations have showcased predictive efficacy in examining the role of SNPs in CIPN. For instance, individuals receiving oxaliplatin therapy exhibited five discerned SNPs, foretelling the onset of CIPN with an accuracy rate of 72% [103]. Other researchers expanded upon these predictive discoveries to encompass further polymorphisms discovered in genes linked to Charcot-Marie-Tooth disease [104]. The genes that exhibited significant associations with CIPN were connected to myelinating Schwann cells (periaxin), the velocity of nerve conduction (Rho guanine nucleotide exchange factor 10), and the synthesis of tachykinin peptides. Some studies have suggested that genetics may influence the risk of developing CIPN. For instance, a genome-wide association study (GWAS) on CIPN from paclitaxel, carboplatin, and oxaliplatin found a genetic variant near the PDE6C gene that was strongly associated with CIPN [105]. Another GWAS on CIPN from different drugs, including bortezomib and vincristine, identified several genetic variants near genes related to nerve function, such as FBXO33, INTU, BCL6, and IL17RB [106]. However, the genetic predictors of CIPN are not well established, and the results of different studies may not be consistent or replicable. Therefore, more research is needed to confirm the role of genetics in CIPN and to elucidate how these mutations influence the development of this disorder. Moreover, to develop reliable biomarkers for predicting and preventing CIPN in clinical practice, further studies are necessary.

## 3. Discussion and Conclusions

Chemotherapy remains the main treatment option for different types of cancer, but unfortunately, it causes the complex and disabling nerve disorder called CIPN. CIPN is a common and unbearable side effect affecting the peripheral nervous system that causes symptoms such as pain, numbness, burning, and loss of sensation. CIPN can persist for months or years after the completion of chemotherapy and impair the quality of life of patients [107]. No efficacious therapies exist to treat or counteract this disorder. The use of opioids, often recommended to treat different chronic pain disorders, provides only limited benefit for chemotherapy-induced pain and poses the risk of dependence.

Hence, it is crucial to explore therapeutic alternatives that can offer more effective and sustainable solutions.

The exact mechanisms of CIPN are not fully understood, but several factors have been proposed to contribute to the development and maintenance of CIPN [108]. These include a direct neurotoxicity, since chemotherapeutic agents can damage the structure and function of sensory neurons and their axons, leading to degeneration, apoptosis, or altered excitability (Figure 1). Chemotherapeutics can interfere also with the integrity of organelles, such as lysosomes and mitochondria. Platinum compounds form DNA adducts that trigger DNA damage response and oxidative stress in neurons [109].

Another mechanism involved in CIPN onset and maintenance is the inflammatory response at peripheral and central nervous system levels, concerning the activation of glial cells, the release of pro-inflammatory cytokines and chemokines, and the infiltration of immune cells. These inflammatory mediators can activate and sensitize nociceptors, increase synaptic transmission, and modify ion channels and receptors involved in pain signaling [110]. Consequently, ion channels such as TRPA1, TRPV1, TRPV4, and TRPM8 are implicated in CIPN, as they can be activated or modulated by chemotherapeutic agents or inflammatory mediators. TRP channel activation can improve the sensitivity and excitability of sensory neurons and participate in pain perception. Another relevant mechanism involved in CIPN is the NR2B receptor, a glutamate receptor upregulated in the spinal cord and the brain after chemotherapy, and its activation can enhance the pain signal and induce central sensitization [13].

These mechanisms are not mutually restricted, and they may interact and influence each other in intricate ways (Figure 2). Hence, CIPN is a multifactorial and heterogeneous disorder that requires a complete and individualized approach for prevention and treatment. Currently, there are no effective drugs for CIPN, but some promising targets and strategies have been identified, such as chemokine antagonists, TRP channel blockers, NR2B antagonists, and SARM1 inhibitors. 

In particular, SARM1 inhibitors, a new category of drugs, are garnering increasing interest in the treatment of CIPN. SARM1, also known as Sterile Alpha and TIR Motif-Containing 1, is a protein involved in the degeneration of nerve fibers and neuronal cell death. It is believed that inhibition of SARM1 in the context of CIPN may protect peripheral nerves by reducing the risk of neuropathic damage and alleviating symptoms associated with neuropathy. These novel drugs represent a promising therapeutic approach to prevent or mitigate the severity of CIPN and enhance the well-being of patients undergoing chemotherapy [111].

The results of the studies reported in this review are very promising; however, more investigations are necessary to elucidate the molecular and cellular mechanisms of CIPN and to develop novel, safer, more targeted, and effective therapies for this challenging complication of chemotherapy.

## Figures and Tables

**Figure 1 biomedicines-12-00751-f001:**
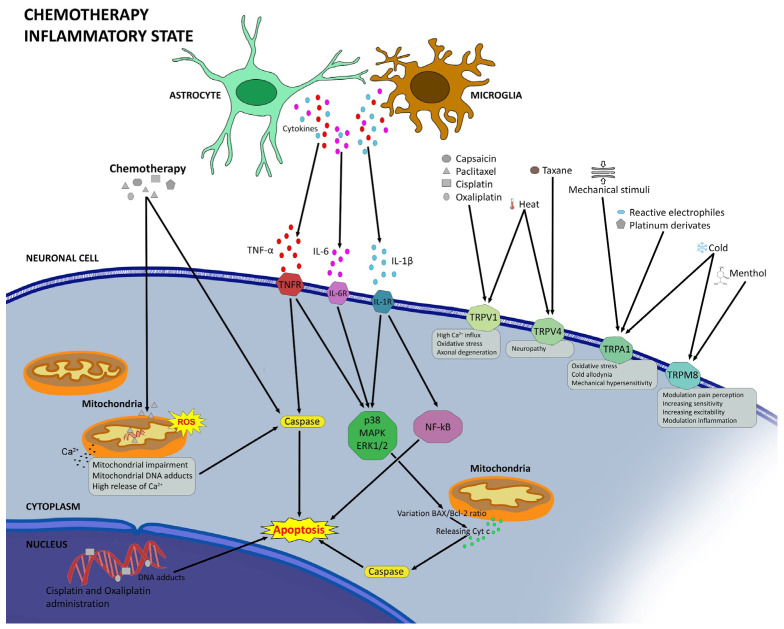
Schematic figure of main mechanisms of CIPN.

**Figure 2 biomedicines-12-00751-f002:**
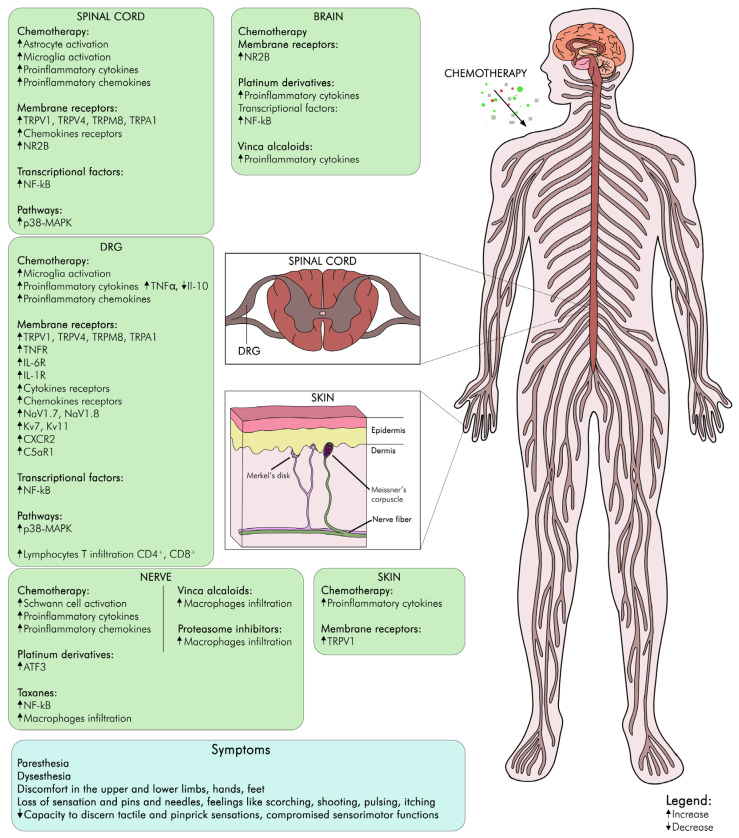
Schematic figure of main actors involved in CIPN.

## Data Availability

Not applicable.

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
