# Peer review of "Molecular and Cellular Involvement in CIPN"

_biomedicines, 2024, doi:10.3390/biomedicines12040751_

Round 1

Reviewer 1 Report

Comments and Suggestions for Authors

The paper does not present any particular errors but does not bring anything new in the field of peripheral neurotoxicity induced by chemotherapy. The bibliography is not particularly updated. I suggest the authors focus on one of the points addressed, for example neuroinflammation in CIPN.

 In particular:

Lines 184-186 The statement "At DRG..........nerve damage [47, 48].I do not understand authors mention ATF-3 in this context.

Linee 272-280 Authors discuss about ALCAR. Studies are very old and not worth mentioning.

Lines 289-298 Authors discuss the effect of some antineplastic drugs on microtubules. This topic deserve a dedicated paragraph.

Moreover authors do not cite the article DOI: 10.1038/s41598-021-89856-3 in which authors demonstrate for the first time a direct interaction between Bortezomib and microtubule.

Lines 328-332 The statement "Likewise, .......in rats [89]" is not clear. The suppression of CCL2 prevents the reduction of density of IENSs.  The statement says the opposite.

Authors mention SARM1 without discussing its role which recently appears increasingly central in axonal degeneration.

Author Response

“The paper does not present any particular errors but does not bring anything new in the field of peripheral neurotoxicity induced by chemotherapy. The bibliography is not particularly updated. I suggest the authors focus on one of the points addressed, for example neuroinflammation in CIPN.”

RESPONSE: We would like to thank Reviewer 1 for the comments provided that greatly improved our manuscript. We did our best to address all the points raised and we now updated the bibliography as suggested.

" Lines 184-186 The statement "At DRG..........nerve damage [47, 48].I do not understand authors mention ATF-3 in this context."

RESPONSE: we appreciate the Reviewer’s comment. We agree with the reviewer and we now briefly explained its role in the context of CIPN.

“Lines 272-280 The authors discuss ALCAR. The studies are very old and not worth mentioning.”

RESPONSE: thank you for the comment and suggestion. After conducting further research on ALCAR's involvement in CIPN, we agree with the reviewer that the studies are old and not worth mentioning.

“Lines 289-298 Authors discuss the effect of some antineplastic drugs on microtubules. This topic deserve a dedicated paragraph.”

RESPONSE: thank you for the comment. A discussion on the effect of some antineoplastic drugs on microtubules has been added at the end of the paragraph.

“Moreover authors do not cite the article DOI: 10.1038/s41598-021-89856-3 in which authors demonstrate for the first time a direct interaction between Bortezomib and microtubule.”

RESPONSE: we apologize for not mentioning the article where the authors demonstrated for the first time a direct interaction between Bortezomib and microtubules. It has been added to our review as suggested.

“Lines 328-332 The statement "Likewise, .......in rats [89]" is not clear. The suppression of CCL2 prevents the reduction of density of IENSs.  The statement says the opposite.”

RESPONSE: we appreciate the Reviewer’s comment. We have rephrased the sentence to make it clearer.

“Authors mention SARM1 without discussing its role which recently appears increasingly central in axonal degeneration”.

RESPONSE: Thank you for the comment. We have now added a brief description of the role of SARM1 inhibitors as a potential treatment for CIPN.

Reviewer 2 Report

Comments and Suggestions for Authors

The number of patients treated with anti-cancer drugs are increasing in the world. Accordingly, CIPN became the critical problem in the patients with cancer. This manuscript well reviews the current knowledge and challenges in the field of CIPN, especially focusing on the biological and molecular mechanisms underlying CIPN.

  I think that this manuscript has some value to be published in Biomedicines to understand the molecular mechanisms underlying CIPN and develop the new therapy and prevention of the disease.

Comments to the authors. 

Major comments: 

  Authors focus on molecular and cellular mechanism of CINP in this review. I think that it would be better to address some comments about “Nonpharmacological therapy” (Ex. Biomed Pharmacother. 2022 Mar:147:112671. doi: 10.1016/j.biopha.2022.112671. Epub 2022 Jan 29. Treatment and diagnosis of chemotherapy-induced peripheral neuropathy: An update. Allison D Desforges et al.), “Molecular Targeted Therapies”, and “monoclonal antibodies (mAbs) for cancer therapy”.

Minor comments:  

L43~44; CIPN throughout their treatment [13] regimen.

> CIPN throughout their treatment regimen [13].

Authors repeat same sentence with different references s follows;

L60; Approximately 68% of patients encounter CIPN post-chemotherapy, with its persistence observed in roughly one-third of patients beyond the 6-month mark [12].

L63~64: Approximately 68% of patients encounter CIPN post-chemotherapy, with its persistence observed in roughly one-third of patients beyond the 6-month mark [15].

Author Response

The number of patients treated with anti-cancer drugs are increasing in the world. Accordingly, CIPN became the critical problem in the patients with cancer. This manuscript well reviews the current knowledge and challenges in the field of CIPN, especially focusing on the biological and molecular mechanisms underlying CIPN.

I think that this manuscript has some value to be published in Biomedicines to understand the molecular mechanisms underlying CIPN and develop the new therapy and prevention of the disease”.

RESPONSE: We appreciate Reviewer 2 positive comments for the time spend in revised our manuscript, we did our best to address all the points raised.

Major comments: 

“Authors focus on molecular and cellular mechanism of CINP in this review. I think that it would be better to address some comments about “Nonpharmacological therapy” (Ex. Biomed Pharmacother. 2022 Mar:147:112671. doi: 10.1016/j.biopha.2022.112671. Epub 2022 Jan 29. Treatment and diagnosis of chemotherapy-induced peripheral neuropathy: An update. Allison D Desforges et al.), “Molecular Targeted Therapies”, and “monoclonal antibodies (mAbs) for cancer therapy”.”

RESPONSE: We thank the reviewers for the comment. We now discussed about this point as suggested (please see lines 68-82)

Minor comments:  

“L43~44; CIPN throughout their treatment [13] regimen.

> CIPN throughout their treatment regimen [13].”

RESPONSE: We apologize for the oversight; we corrected this point.

“Authors repeat same sentence with different references s follows;

L60; Approximately 68% of patients encounter CIPN post-chemotherapy, with its persistence observed in roughly one-third of patients beyond the 6-month mark [12].

L63~64: Approximately 68% of patients encounter CIPN post-chemotherapy, with its persistence observed in roughly one-third of patients beyond the 6-month mark [15].”

RESPONSE: We apologize for the oversight; we modified the references as suggested.

We would like to thank again the reviewers and editors for the time spent in revising our manuscript. We hope that it is now suitable for publication.

Round 2

Reviewer 1 Report

Comments and Suggestions for Authors

Authors reply to al my comments by correcting the text.

Despite this, paragraph dedicated to microtubules is still not sufficently clear and presents some conceptual errors:

line 337 "primarily operating....." is not correct for Oxaliplatin and Bortezomib

line 345 "This process leads to microtubule destabilization and subsequent cell cycle arrest". Are authors sure of this statement? Is Microtubule destabilization a consequence of DNA damage induced by oxaliplatin?

lines 348-350 Authors discuss about effect of vinca derivatives in cycling cells not in neurons

Author Response

Reviewer 1:

“Line 337 "primarily operating....." is not correct for Oxaliplatin and Bortezomib.”

RESPONSE: Thank you for the comment, we have corrected the sentence.

" Line 345 This process leads to microtubule destabilization and subsequent cell cycle arrest. Are authors sure of this statement? Is Microtubule destabilization a consequence of DNA damage induced by oxaliplatin?"

RESPONSE: Thank you for your feedback. We have refined the concept to enhance its clarity and comprehension.

“Lines 348-350 Authors discuss about effect of vinca derivatives in cycling cells not in neurons.”

RESPONSE:  Thank you for your observation. We have supplemented the content with insights concerning the impact of vinca alkaloid derivatives on neuronal function.

Round 3

Reviewer 1 Report

Comments and Suggestions for Authors

The manuscript uploaded is not the correct version but the old one. I read the new version received by the office.

 Authors reply to all my comments and correct manuscript. Remain a conceptual error in the statement " Bortezomib, a proteasome inhibitor, primarily exerts its effect through indirect action on microtubule stability" at lines 343-345.  

Bortezomib is primarily a proteasome inhibitor and its antineoplastic effect is probably primarily due to the block of proteasome. 

Malacrida et al. have demonstrated for the first time the direct interaction  of Bortezomib with microtubule and this suggest that the neurotoxic effect could be due to this interaction

Author Response

Reviewer 1:

“Authors reply to all my comments and correct manuscript. Remain a conceptual error in the statement " Bortezomib, a proteasome inhibitor, primarily exerts its effect through indirect action on microtubule stability" at lines 343-345. 

Bortezomib is primarily a proteasome inhibitor and its antineoplastic effect is probably primarily due to the block of proteasome.

Malacrida et al. have demonstrated for the first time the direct interaction  of Bortezomib with microtubule and this suggest that the neurotoxic effect could be due to this interaction.”

RESPONSE: Thank you for your feedback. We have made adjustments to the sentence to improve its clarity and understanding.

We would like to thank again the reviewers and editors for the time spent in revising our manuscript. We hope that it is now suitable for publication.
